# C1 CAGE detects transcription start sites and enhancer activity at single-cell resolution

Tsukasa Kouno[1], Jonathan Moody [1], Andrew Tae-Jun Kwon [1], Youtaro Shibayama[1], Sachi Kato[1], Yi Huang[1,3], Michael Böttcher[1], Efthymios Motakis[1,4], Mickaël Mendez [1,5], Jessica Severin [1], Joachim Luginbühl[1], Imad Abugessaisa [1], Akira Hasegawa[1], Satoshi Takizawa[1], Takahiro Arakawa[1], Masaaki Furuno[1], Naveen Ramalingam [2], Jay West[2], Harukazu Suzuki [1], Takeya Kasukawa [1], Timo Lassmann[1,6], Chung-Chau Hon[1], Erik Arner[1], Piero Carninci [1], Charles Plessy[1,7] & Jay W. Shin[1]

Single-cell transcriptomic profiling is a powerful tool to explore cellular heterogeneity. However, most of these methods focus on the 3′-end of polyadenylated transcripts and provide only a partial view of the transcriptome. We introduce C1 CAGE, a method for the detection of transcript 5′-ends with an original sample multiplexing strategy in the C1[TM] microfluidic system. We first quantifiy the performance of C1 CAGE and find it as accurate and sensitive as other methods in the C1 system. We then use it to profile promoter and enhancer activities in the cellular response to TGF-β of lung cancer cells and discover sub-populations of cells differing in their response. We also describe enhancer RNA dynamics revealing transcriptional bursts in subsets of cells with transcripts arising from either strand in a mutually exclusive manner, validated using single molecule fluorescence in situ hybridization.

[1] RIKEN Center for Integrative Medical Sciences (IMS), 1-7-22 Suehiro-cho, Tsurumi-ku, Yokohama 230-0045, Japan. [2] Single-Cell Research and Development, Fluidigm Corporation, 7000 Shoreline Court, Suite 100, South San Francisco 94080 CA, USA. [3] Present address: ACT Genomics Co. Ltd., 3F., No. 345, Xinhu 2nd Rd, Neihu Dist., Taipei City 114, Taiwan. [4] Present address: Yong Loo Lin School of Medicine MD6, #08-01, 14 Medical Drive, National University of Singapore, Singapore 117599, Singapore. [5] Present address: Princess Margaret Cancer Research Tower 11-401, 101 College Street, Toronto, ON M5G 1L7, Canada. [6] Present address: Telethon Kids Institute, The University of Western Australia, Perth Children's Hospital, 15 Hospital Ave, Nedlands 6009 WA, Australia. [7] Present address: Okinawa Institute of Science and Technology Graduate University (OIST), 1919-1 Tancha, Onna-son, Kunigami-gun, Okinawa 904-0495, Japan. These authors contributed equally: Tsukasa Kouno, Jonathan Moody, Andrew Tae-Jun Kwon. Correspondence and requests for materials should be addressed to C.P. (email: charles.plessy@oist.jp) or to J.W.S. (email: jay.shin@riken.jp)

Single-cell transcriptomic profiling can be used to uncover the dynamics of cellular states and gene regulatory networks within a cell population[1,2]. Most available single-cell methods capture the 3′-end of transcripts and are unable to identify where transcription initiates. Instead, capturing the 5′-end of transcripts allows the identification of transcription start sites (TSS) and thus the inference of the activities of their regulatory elements. Cap analysis gene expression (CAGE), which captures the 5′-end of transcripts, is a powerful tool to identify TSS at single-nucleotide resolution[3,4]. Using this technique, the FANTOM consortium has built an atlas of TSS across major human cell-types and tissues[5], analysis of which has led to the identification of promoters as well as enhancers in the human genome[6,7]. Enhancers have been implicated in a variety of biological processes[8,9], including the initial activation of responses to stimuli[10] and chromatin remodeling for transcriptional activation[11]. In addition, over 60% of the fine-mapped causal non-coding variants in autoimmune disease lay within immune-cell enhancers[12], suggesting the relevance of enhancers in pathogenesis of complex diseases. Enhancers have been identified by the presence of balanced bidirectional transcription producing enhancer RNAs (eRNAs), which are generally short, unstable and non-polyadenylated (non-polyA)[6]. Single-molecule fluorescence in situ hybridization (smFISH) studies have suggested that eRNAs are induced with similar kinetics to their target mRNAs but that co-expression at individual alleles was infrequent[13]. However, the majority of enhancer studies have been conducted using bulk populations of cells meaning that the dynamics of how multiple enhancers combine to influence gene expression remains unknown.

The majority of single-cell transcriptomic profiling methods[14] rely on oligo-dT priming during reverse-transcription, which does not capture non-polyA RNAs transcripts (e.g., eRNAs). The recently developed RamDA-seq[15] method uses random priming to capture the full-length non-polyA transcripts including eRNAs. However, this method is not strand-specific and unable to pinpoint transcript 5′-ends; thus, it cannot detect the bidirectionality of eRNA transcription and it is difficult to distinguish reads derived from the primary transcripts of their host gene (i.e., intronic eRNAs). Methods are typically implemented for a specific single-cell handling platform (e.g., microwell, microfluidics, or droplet-based platforms)[14], because each platform imposes strong design constraints on the critical steps of cell lysis and nucleic acid handling. The proprietary C1™ Single-Cell Auto Prep System (Fluidigm) uses disposable integrated fluidic circuits (IFCs) and provides a registry of publicly available single-cell transcriptomics methods (Supplementary Table 1), which can be customized. Previously, we introduced nano-CAGE[16], a method requiring only nanograms of total RNA as starting material, based on a template switch mechanism combined with random priming to capture the 5′-ends of transcripts independent of polyA tails in a strand-specific manner. Here, we develop C1 CAGE, a modified version of nano-CAGE customized to the C1 system to capture the 5′-ends of transcripts at single-cell resolution.

Current single-cell methods are usually limited in the number of samples that can be multiplexed within the same run. Thus, experimental designs requiring multiple replicates and different conditions are prone to batch effects, confounding biological information with the technical variation of each experiment[17]. To mitigate batch effects, we took advantage of the transparency of the C1 system to encode multiple perturbation states in a single run by fluorescent labeling and imaging.

We apply this method to investigate the response to TGF-β in A549 cells, an adenocarcinomic human alveolar basal epithelial cell line. TGF-β signaling plays a key role in embryonic development, cancer progression, host tumor interactions, and driving epithelial-to-mesenchymal transition (EMT)[18,19]. We examine the response to TGF-β in A549 cells to uncover dynamically regulated promoters and enhancers at single-cell resolution. We observe an asynchronous cellular response to TGF-β in subpopulations of cells. We also investigate the dynamics of enhancer transcription at single-cell resolution with validation by smFISH. Our results suggest transcriptional bursting of enhancers as reflected by high expression of eRNAs in a few cells. Also, while in pooled cells enhancers show bidirectional transcription, within single-cells transcription at enhancers is generally uni-directional—i.e., transcription on the two strands seems to be mutually exclusive.

## Results

**Development of C1 CAGE.** We developed the C1 CAGE method, based on nano-CAGE[16], C1 STRT Seq[20], and C1 RNA-seq[21], implementing reverse-transcription with random hexamers followed by template switching and pre-amplification (Fig. 1a). The cDNA is tagmented and the 5′-end of cDNA is specifically amplified by index polymerase chain reaction (PCR). The resulting library is sequenced from both ends, with the forward reads identifying the 5′-end of the transcript at single-nucleotide resolution and the reverse read identifying downstream regions of the matching transcript.

To assess the ability of C1 CAGE to detect differential expression we compared libraries prepared using two reference mixtures of synthetic spike-in molecules—sets of 92 exogenous control transcripts with defined abundances developed by the External RNA Controls Consortium (ERCC)[22]—with fixed ratios of input amounts at 4-, 1-, 2/3-, and 1/2-fold difference. Fitting a linear model we find an $R$-squared value of 87% (Fig. 1b). Next, to assess the specificity of 5′-end capture, we analyzed the positions of forward reads on these spike-ins and found that ~80% of their 5′-ends align to the first base (Fig. 1c), supporting the specificity of 5′-end capture in C1 CAGE. Of the remaining (~20%) reads, half of them can be explained by "strand-invasion" events, which are artefacts arising from interruption of first-strand synthesis due to complementarity with the template switching oligonucleotide. Strand invasion is present in methods using template switching including C1 STRT[23] (Supplementary Fig. 1) and can be identified based on the upstream sequence of the read. Next, we compared the quantification accuracy, molecular detection limit[24] and number of genes detected by C1 CAGE and C1 STRT. For quantification accuracy, measured as the Pearson correlation between the input spike-in amounts and the observed read counts, C1 CAGE displayed a median of 0.79, slightly higher (Welch's two sample $t$ test, two-sided: $t = 4$, d$f = 127.6$, $p < 0.0001$) than C1 STRT (median of 0.74, Fig. 1d, S1b, c). For detection limit, measured as the median number of spike-in molecules required to give a 50% chance of detection, C1 CAGE displayed a median of 22, which is significantly more sensitive (Welch's two sample $t$ test, two-sided: $t = -14$, d$f = 94.2$, $p < 2.2e-16$) compared with C1 STRT (median of 146, Fig. 1e). Finally, to compare the number of protein-coding genes detected in a biological sample we prepared a C1 CAGE library in mouse embryonic stem cells (MES) in order to compare with an existing C1 STRT dataset[24]. Processing 500,000 downsampled reads from both datasets, counting within FANTOM5 promoter regions, C1 CAGE detects median 2991 protein-coding genes compared to median 2335 in C1 STRT (Fig. 1f, Welch's two sample $t$ test, two-sided: $t = 7.033$, d$f = 128.1$, $p < 1e-5$). These results demonstrate that C1 CAGE specifically captures the 5′-end of transcripts, detects differential expression with high accuracy, has quantification accuracy and detection sensitivity comparable to other C1-system methods.

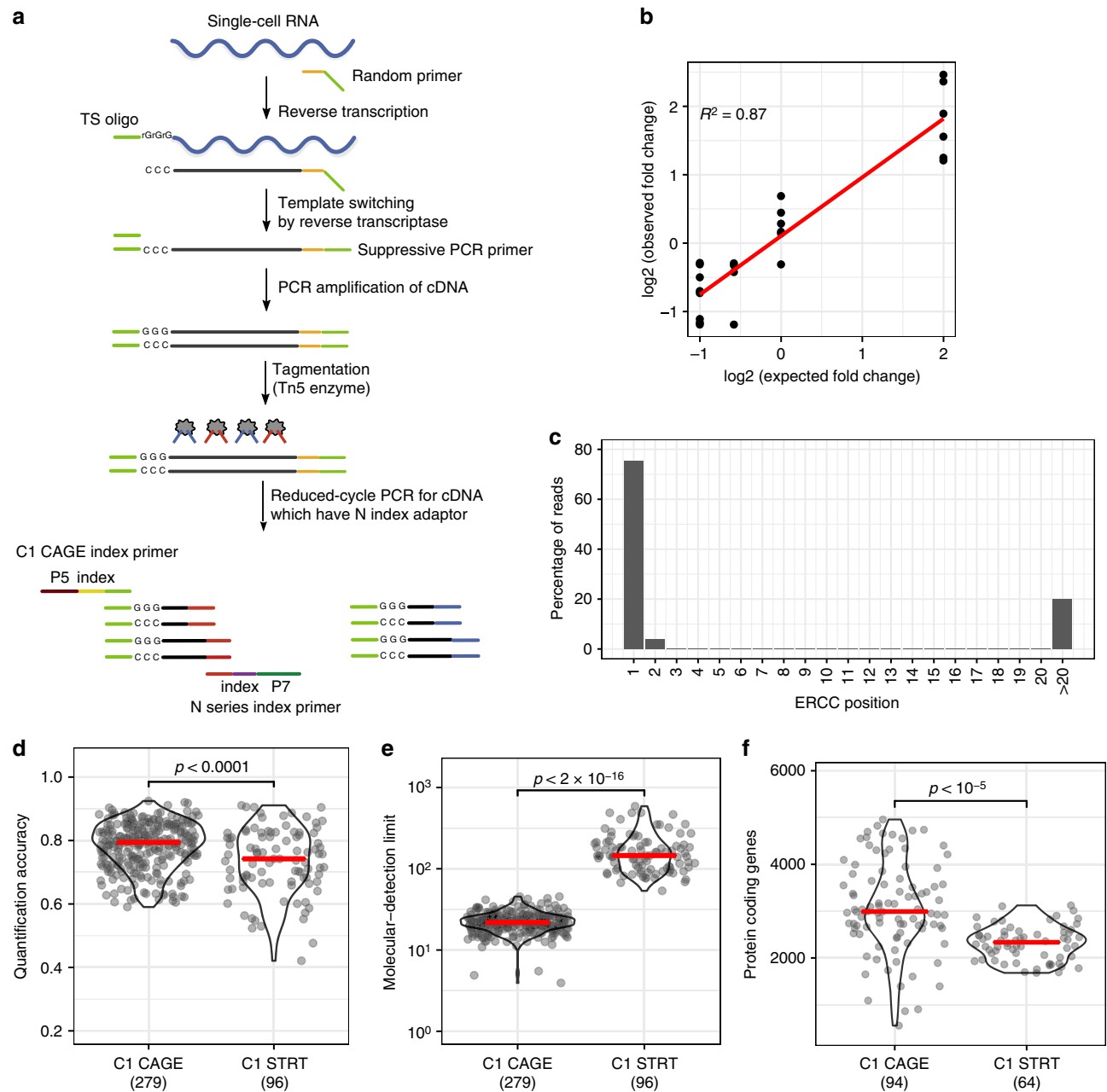

**Fig. 1** The C1 CAGE method and performance. **a** Schematic of the C1 CAGE method. Tn5 enzymes are loaded with two different adapters: N (red) and S (blue). P5, P7: Illumina sequencing adapters. **b** Observed and expected fold-change ratios between ERCC mix1 and mix2. Linear regression line (red) and R-squared value shown. **c** Percentage of reads aligning to the 5′-end of ERCC spike-ins by nucleotide position. **d–f** Comparison between C1 CAGE and C1 STRT Seq (data from Svensson et al.[24]) Red bars show median values. *p*-values from Welch's two-sided two sample *t* test shown. **d** Pearson correlation between expected and observed ERCC spike-in molecules. **e** The number of ERCC spike-in molecules required for a 50% chance of detection. **f** Protein-coding genes detected in mouse ES cells counting reads within FANTOM5 promoter regions. Source data are provided as a Source Data file

**Color multiplexing**. Taking advantage of the imaging capacities of the C1 system, we devised a strategy to multiplex samples within the same C1 CAGE replicate, by labeling cells with different Calcein AM dyes to both encode sample information and monitor cell viability[25]. We observed no or minor responses to these dyes in qPCR of known TGF-β-induced genes (Supplementary Fig. 2). To validate this strategy, we combined differently labeled mouse embryonic fibroblasts (MEF) and human dermal fibroblasts (HDF) in the same C1 CAGE run; demultiplexing correctly separated cells by species in agreement with mapping rates to a combined human and mouse genome (Supplementary

Fig. 3). Based on this approach, we multiplexed samples of A549 cells stimulated with TGF-β in a time-course at three time points (0, 6, and 24 h, in triplicates) by permuting the Calcein green and red AM dyes for each time point in each replicate (Fig. 2a). The three C1 CAGE replicates were sequenced to a median depth of 2.4 million raw read pairs per cell. Analyzing the genomic distribution of forward read 5′-ends per replicate, a mean of 34% and 0.7% of reads were aligned to promoter and enhancer CAGE clusters, respectively (Fig. 2b). Subsampling analysis demonstrates the number of CAGE clusters detected in most single-cells are saturated at the current sequencing depths, with a median of 2788

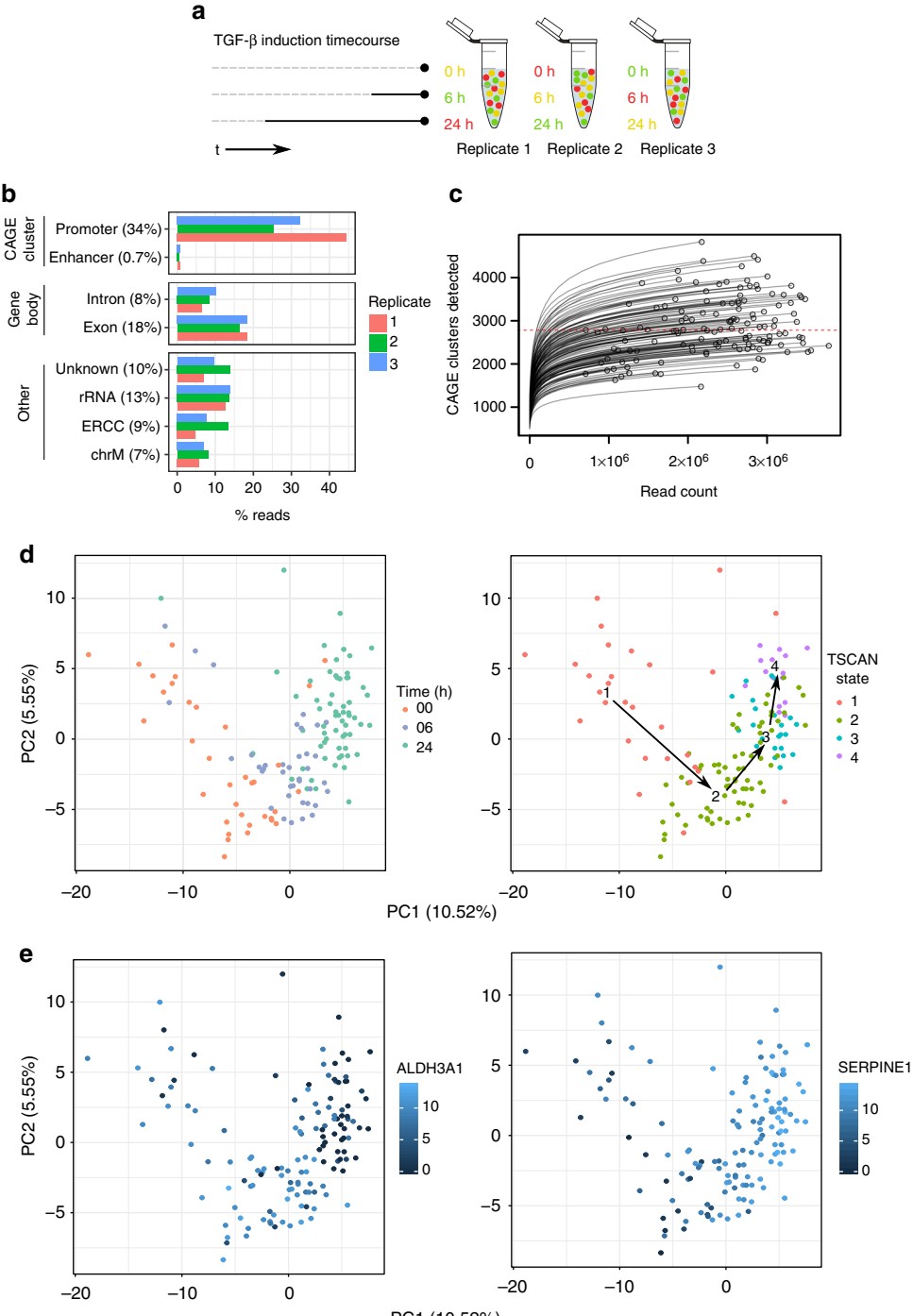

**Fig. 2** Multiplexing time course strategy. **a** Different combinations of Calcein red and green are added to each timepoint for each replicate. **b** Forward read 5′-end counts by annotation category. Mean read percentage per category shown in brackets. **c** Count of CAGE clusters within each cell after subsampling. Dashed red line at median (2788). **d** PCA of cells performed on variable subset of CAGE clusters, percentage of variance explained by components shown, cells colored by time point and TSCAN state. **e** PCA of cells performed on variable subset of CAGE clusters, percentage of variance explained by components shown, cells colored by expression values for the marker genes *ALDH3A1* and *SERPINE1* demonstrating that the dynamics of TGF-β response are captured by the TSCAN states. Source data are provided as a Source Data file

CAGE clusters detected per cell (Fig. 2c). To demultiplex time points, we localized the cells in their capture chambers on the IFCs and quantified their fluorescence in the red and green channels, identifying 40, 41, and 70 cells for time points 0, 6, and 24 h, respectively. Following the scran pipeline[26] we removed 15 unreliable cells, arriving at the final set of 136 high-quality cells. Initially, we observed a strong batch effect with principal

components analysis (PCA), where cells cluster by replicate (Supplementary Fig. 4a). However, our cell labeling design ensured that each replicate contained cells for each time point, allowing us to correct for this batch effect using linear modeling. After batch correction cells were clustered by time points rather than by replicate (Supplementary Fig. 4b). After removing low abundance CAGE clusters, our final dataset detected 18,687

CAGE clusters, covering 9809 GENCODE genes (Supplementary Fig. 5; annotation breakdown) and 826 FANTOM5 enhancers. All subsequent analyses are based on this normalized and batch-corrected expression table in log2 scale (Supplementary Data 1). For comparison, we generated corresponding bulk CAGE data using the nAnT-iCAGE method[27] for each sample (0, 6, and 24 h, in triplicates) sequenced to median a depth of 10.7 M reads. Bulk CAGE and C1 CAGE libraries similar distributions of reads along protein-coding genes (Supplementary Fig. 6a, b) with median 2268 protein-coding genes detected per single cell while bulk CAGE libraries detect ~13,000 (Supplementary Fig. 6c, d).

**Dynamic TSS regulation upon TGF-β treatment.** To identify transcription start sites (TSSs) that are dynamically regulated during TGF-β treatment, we performed pseudotime analysis on a variable subset of CAGE clusters with TSCAN[28]. TSCAN divided the pseudotime ordering into four distinct states, which showed considerable consistency with the time points, as seen by PCA (Fig. 2d). We also confirmed the consistency of the TSCAN states by visualizing the expression levels of two highly variable CAGE clusters for known EMT marker genes, *ALDH3A1* and *SERPINE1*, which showed a clear shift in expression levels from 0 to 24 h (Fig. 2e). To understand the influence of the cell cycle on how TSCAN defined the states, we calculated G2M scores with the *cyclone* package using the pre-calculated data trained on human embryonic stem cells (hESCs)[29,30]. The clear separation of scores between states 1 and 2 points to the possibility that half (16/35) of 0 h cells were in proliferative states prior to TGF-β stimulation (Fig. 2d and Supplementary Fig. 7).

Next, to identify genes that are co-regulated across the TSCAN states, we performed weighted gene co-expression network analysis (WGCNA)[31], correlating CAGE cluster expression levels across cells. We visualized their trajectories across the pseudotime using eigengene profiles to represent the average behavior and highlight two examples from each module with eigengene correlation coefficient of at least 0.3 with *p* value less than 0.1 (Fig. 3a, b). The module labels were assigned based on these trajectory visualizations: suppressed ($n = 1041$), early ($n = 1775$), and late responders ($n = 2223$) representing those genes that undergo strong expression changes with TGF-β activation, whereas weak responding I ($n = 825$) and II ($n = 164$) represent those with little or no changes in their transcription.

To understand the biological contexts of these modules, we investigated the enrichment of transcription factor binding motifs[32,33] and Gene Ontology (GO) terms in each module. Examining motifs enriched in all modules against a randomly generated GC-matched background, we find that the ETS-related genes, a well-defined family of transcription factors known to promote metastasis progression in EMT are most prominent[34] (Supplementary Fig. 8).

Examining each module individually against the combined background of all the other modules (Fig. 3c, d) we observe the suppressed module enriched in GO terms related to DNA replication and cell cycle. It has been reported that early after TGF-β treatment, the expression of multiple genes that play key roles in regulating cell cycle progression are suppressed[35]. We observe suppressed expression of *CCNB2* known to interact with the TGF-β pathway in promoting cell cycle arrest[36] and of *ALDH3A1* known to affect cell growth in A549 cells[37]. We also observe enriched motifs for the cell cycle regulators *LIN54* and *GFI1*[38,39]. CAGE clusters in the suppressed module are more highly expressed in TSCAN state 1, which may represent cells that have not yet fully undergone TGF-β induced G1 arrest as explained above.

Within the early responders and late responders modules we observe canonical TGF-β response genes, including *KLF6* known to suppress growth through TGF-β transactivation[40] and marker genes for EMT such as *SERPINE1* and *FASN*. TGF-β is one of the key signal transduction pathways leading to EMT and several lines of evidence implicate increased TGF-β signaling as a key effector of EMT in cancer progression and metastasis[18,19,41]. We observed upregulation of mesenchymal marker genes, with a clear increase in *Vimentin* (*VIM*) expression starting during TSCAN state 2, and expression of *N-cadherin* (*CDH2*) not detected until TSCAN state 2, and then expressed within a subset of cells (Supplementary Fig. 9).

Within the late responders module we observe enrichment for TFAP2 transcription factors motifs (Fig. 3c). We examined their expression profiles in both the single-cell and bulk data, and found *TFAP2C* to have a strong time-dependent expression profile in bulk data, and sporadic expression in TSCAN states 1 and 2 but absent in the later states (Supplementary Fig. 10). Interestingly, *TFAP2C* is a known marker gene in breast cancer biology, its loss resulting in increased expression of mesenchymal markers associated with the transition from luminal to basal subtypes[42] and the direct repression of cell cycle regulator *CDKN1A*[43,44].

To further dissect the functional heterogeneity in response to TGF-β, we revisited TSCAN states analysis and explored states 3 and 4 which we observe 24 h poststimulation (Fig. 2d). To examine differences between the two states, we performed gene set enrichment analysis amongst CAGE clusters from the early responders and late responders modules with Camera[45] and find a number of gene sets significantly upregulated in TSCAN state 4 including EMT (38 genes, FDR = 0.003; full results in Supplementary Table 2). This suggests the presence of two states in TGF-β 24 h poststimulation response. Interestingly, a previous study implicated the presence of the second state with more severe morphological changes such as cell-to-cell contacts occurring from 10 to 30 h[35]. Thus, the additional states inferred from the pseudotime analysis reveal the asynchronous progression cells upon TGF-β treatment, which would not have been possible with bulk analyses of the three time points.

**eRNA in C1 CAGE.** Next we asked whether C1 CAGE can detect the dynamic expression of eRNAs. We and others have reported that bidirectional transcription is associated with enhancer activity[6]. We observe a similar signature of bidirectional transcription at enhancers detected in pooled C1 CAGE and bulk CAGE data sets (Fig. 4a), as well as a similar enrichment of DNase hypersensitivity and H3K27 acetylation, indicating that C1 CAGE unambiguously detected the transcription of eRNAs at these active enhancer regions (Fig. 4b). To further examine the bidirectionality of eRNAs at a single-cell level, for each enhancer we calculated a bidirectionality score in pooled single-cells ranging from 0 to 1, with 0 being perfectly balanced bidirectional and 1 being perfectly unidirectional. We selected bidirectional enhancers (score < 0.5) with at least 10 reads in at least 5 cells to filter for the most widely and strongly detected enhancers and avoid bias due to dropout. This led to 32 robust bidirectional enhancer loci (Supplementary Fig. 11; the number of enhancers present at various thresholds). However, the bidirectionality scores calculated within single-cells at these loci, were greater than 0.9 (Fig. 4c, shown in detail for one enhancer in Fig. 4d) suggesting that eRNAs are transcribed unidirectionally in single cells, and this directionality was unrelated to timepoint (Supplementary Fig. 12).

Although most enhancers were sporadically detected among single-cells, they were detected at a similar level to promoters in

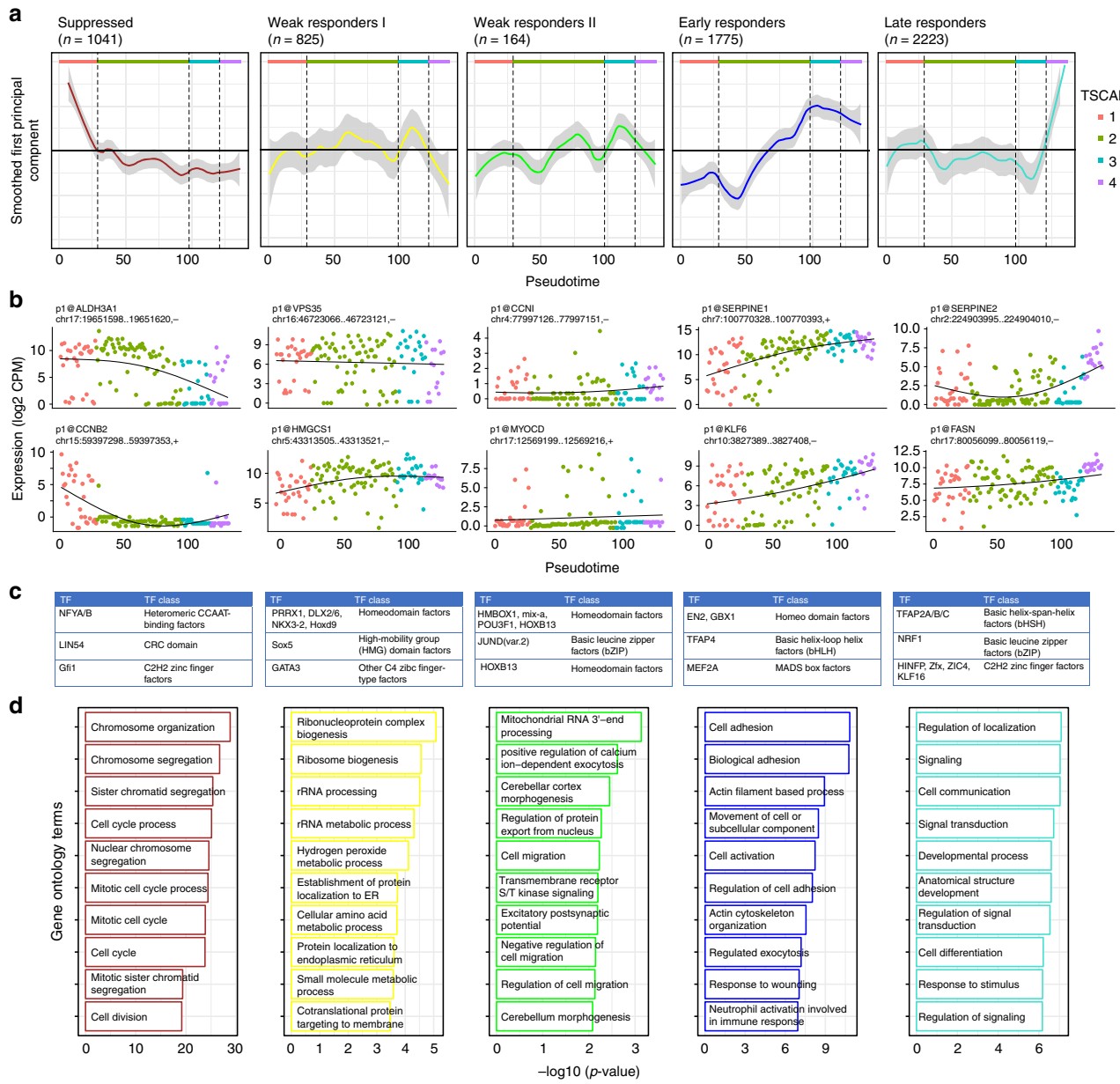

**Fig. 3** WGCNA clusters of response to TGF-β. **a** CAGE cluster expression profiles for 5 WGCNA modules, 3 of which show a clear response to TGF-β (suppressed, early responders, and late responders). The curves are smoothed with the loess R function. **b** Example CAGE cluster expression profiles from each module. **c** Top three enriched TF binding profiles in each module. **d** Functional analysis using edgeR's implementation of GOseq. Top over-represented GO terms for biological processes are shown. Source data are provided as a Source Data file

single-cells when controlling for expression level (Fig. 4e). To assess if enhancers are generally lowly expressed among cells or if they are highly expressed in a subset of cells, we compared the distributions of the maximum expression levels of enhancers and promoters within single-cells and in the bulk data sets (Fig. 4f). While the expression of enhancers is generally lower than that of promoters in the bulk data sets, they have similar distributions of expression levels within single cells. To further evaluate the specificity of enhancer expression in single-cells, we calculated the Gini coefficient for protein-coding promoters and enhancers based on the log of counts. Both sets of features show high Gini coefficients, which may be explained by the sparse nature of single cell expression data; however, enhancers have higher density near the Gini coefficient of 1 (Fig. 4g; Kolmogorov–Smirnov test, $D = 0.38057$, $p$ value < 2.2e−16). This suggests that enhancers behave similarly to promoters, which are

expressed in transcriptional bursts[46,47], but have fewer numbers of cells where bursts of expression take place, which in turn are averaged out by the total population of cells used to obtain the bulk RNA profile.

**FISH validation**. To validate the ability of C1 CAGE to detect eRNAs in single cells, we used smFISH[48,49] to visualize the expression of these transcripts through the TGF-β time course in A549 cells. We first selected intergenic enhancers, filtering out those that overlapped any known transcript models in GENCODEv25, and ranked them by their expression levels. We then searched for their proximal promoters within the same topologically associated domain (TAD) as the potential targets of these enhancers. We selected three enhancers, two of which displayed expression changes across the time-course (Supplementary Figs. 10 and 13) and were

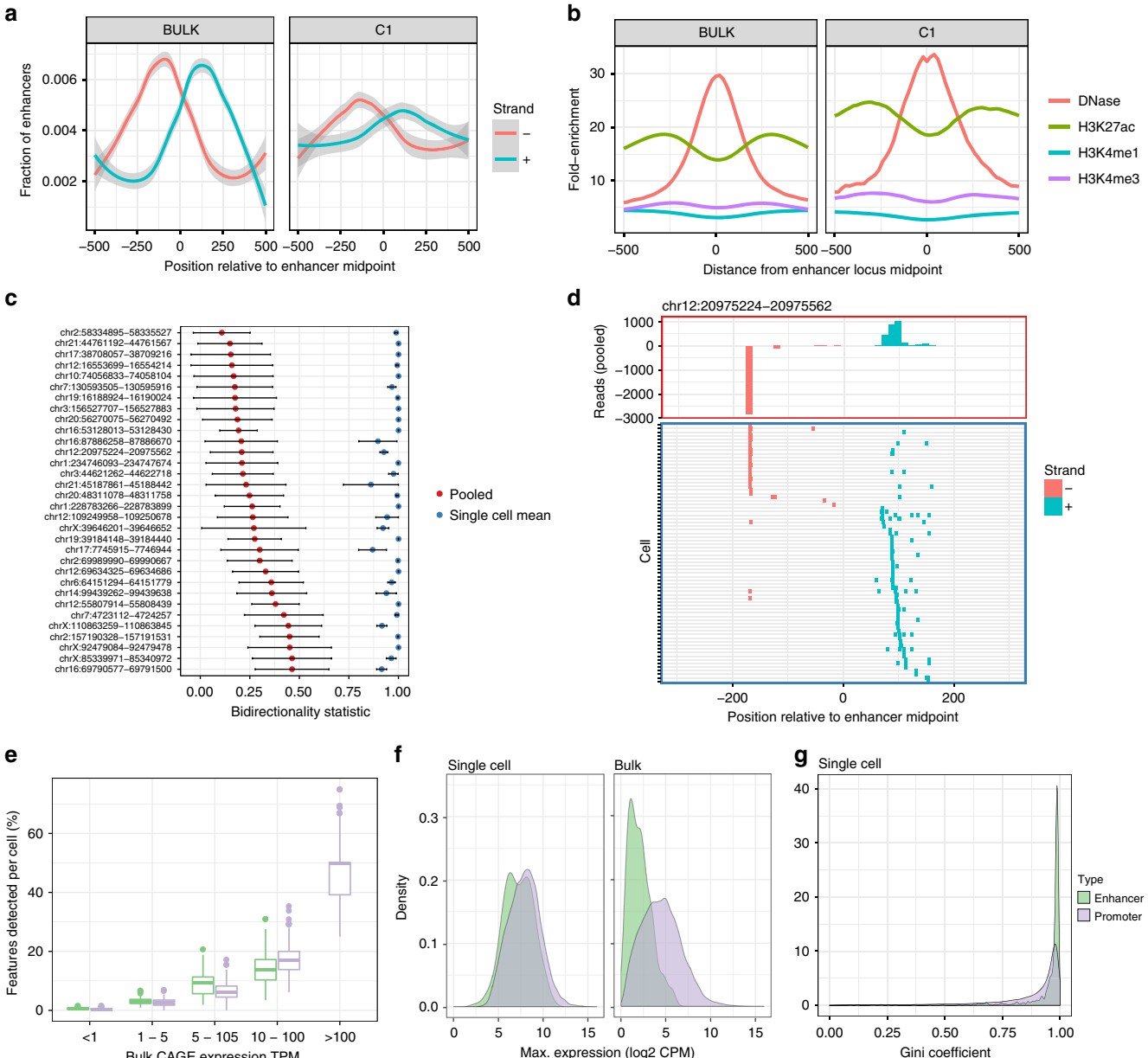

**Fig. 4** Enhancer analysis at single-cell resolution. Comparison of enhancers detected by bulk CAGE and pooled C1 CAGE data **a** showing bidirectional read profiles smoothed by generalized additive model and **b** epigenetic profiles. **c** Bidirectionality scores (0: equally bidirectional; 1: fully unidirectional) at selected enhancers for pooled cells downsampled to the same depth as corresponding single cells 100 times (red dots: mean, black bars: standard error) and single-cells (blue dots: mean; black bars: standard error). **d** Example locus on chromosome 12: read profile histogram (upper box), and read presence or absence in single-cells (lower box). **e–g** Comparison of enhancers and gene promoters in C1 CAGE and bulk CAGE: **e** fraction of bulk features detected within each cell, stratified by bulk expression level, centre line = median, box = 25th to 75th percentiles, whiskers = up to 1.5× interquartile range from the box. **f** Density plots of the maximum expression levels, **g** Gini coefficient distribution in single-cell data. Lower scores: broad expression (expressed in more cells); higher scores: more specific/enriched expression (fewer cells). Source data are provided as a Source Data file

adjacent to genes known to be involved in TGF-β response, *KLF6* and *PMEPA1* (*KLF6*-eRNA1 at chr10:3929991-3930887 and *PMEPA1*-eRNA1 at chr20:56293544-56293843, respectively), and a third enhancer (*PDK2*-eRNA1 at chr17:48105016-48105270) adjacent to *PDK2*.

In line with previous reports[13,50], smFISH for eRNAs gave rise to punctate spots mainly restricted to the nuclei and always no greater than the copy number of the chromosome harboring the enhancer, suggesting that these eRNAs are expressed in low-copy number and remain at or near their site of transcription. Targeting eRNAs on both strands with the same color, smFISH displayed expression profiles similar to C1 CAGE for the *KLF6*-

eRNA1 and *PMEPA1*-eRNA1 enhancers that were upregulated in the C1 CAGE time-course data (Fig. 5a, b). In contrast, *PDK2*-eRNA1, whose expression remained steady in smFISH, decreased in the number of cells with signal across the time course in C1 CAGE (Supplementary Fig. 14).

For validation of our findings that eRNA were expressed unidirectionally within single cells, we also targeted the + and − strands of the *KLF6*-eRNA1 and *PMEPA1*-eRNA1 eRNAs in separate colors. In agreement with the C1 CAGE data for these particular enhancers, the majority of the detected spots belonged to eRNAs from only one strand (Supplementary Fig. 15). In nuclei where eRNAs from both strands were detected, spot

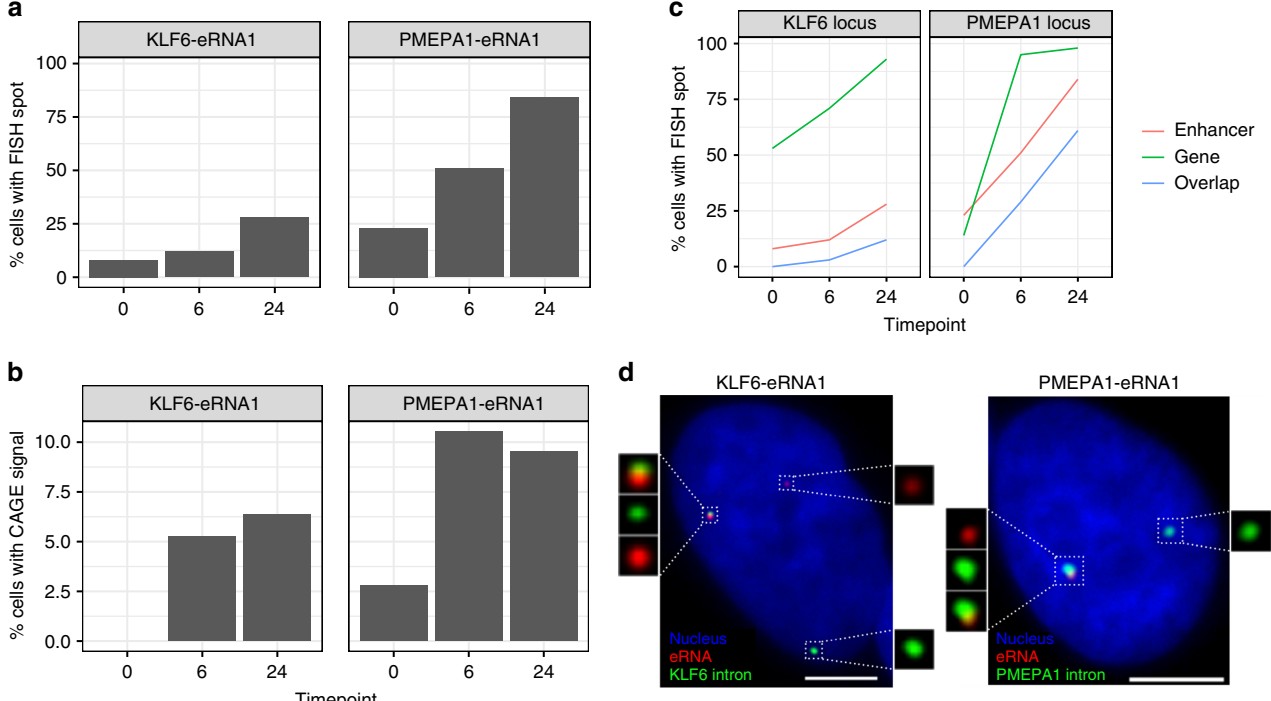

**Fig. 5** Enhancer and promoter profiles in smFISH. **a**, **b** Proportion of cells with KLF6-eRNA1 and PMEPA1-eRNA1 detected by **a** FISH, **b** C1 CAGE. **c** Proportion of cells with detected gene intronic RNA and eRNA and cells with spot overlap at the KLF6 and PMEPA1 loci. **d** Representative images showing gene intronic RNA and eRNA detection by FISH. Bar = 5 μm. n = 100 per time point. Source data are provided as a Source Data file

colocalization was rare (Supplementary Fig. 15), confirming our suggestion that simultaneous bidirectional transcription of enhancers from single alleles is a rare event.

Next, we checked for the association of eRNAs with the transcription of nearby genes using smFISH. Visualization of nearby gene transcription was achieved by targeting only the intronic portion (i.e., nascent RNA). Colocalization of an enhancer RNA spot with a nearby nascent RNA spot would suggest co-expression of the enhancer and the protein-coding gene from the same allele. Interestingly, nascent transcription of nearby protein-coding genes showed similar expression kinetics to the enhancers themselves indicated by increased co-expression of both the protein-coding gene and the nearby eRNA in TGF-β stimulated cells (Fig. 5c, d). For *KLF6*-eRNA1 and *PMEPA1*-eRNA1, we observed time-dependent increase in colocalization and in the number of nuclei with colocalized spots (Fig. 5c, d). In unstimulated cells displaying a basal level of expression of both enhancer and promoter, colocalization of spots could not be observed. This suggests potential stimulus-dependent co-activation of enhancer and its association with the nearby promoter. However, a significant portion of transcription sites expressed no enhancer RNA. Possible reasons include a delayed interval between transcription events from an enhancer and promoter, during which most enhancer RNA is rapidly degraded. It is also possible that other nearby enhancers may exert their effect on a target promoter. In summary, smFISH could validate enhancer expression, including strand specificity, in single-cells as detected by C1 CAGE.

## Discussion
We examined the response to TGF-β in A549 cells to uncover dynamically regulated promoters and enhancers at single-cell resolution. We highlight enhancer dynamics at single-cell resolution and suggest transcriptional bursting of enhancers, and that

while enhancers show bidirectional eRNA transcription in pooled cells, transcripts are generally mutually exclusive.

Among the eight publicly available transcriptome methods for the C1 platform (Supplementary Table 1), only C1 CAGE provides strand-specific whole-transcriptome coverage: its detection of 5′-ends is independent from transcript length and poly-adenylation owing to the use of random primers. To make the method more accessible, we used a commercially available tag-mentation kit in which the transposase is loaded with two different adapters. This adaptation leads to half of the tagmentation products being lost in the process of library preparation. The use of custom loaded transposase, such as in C1 STRT Seq[20], would allow reduction of the final PCR amplification by one cycle and enrich extracted reads in the sequencing library, however at the expense of not using standard reagents.

C1 CAGE has single-nucleotide resolution of transcript 5′-ends, as demonstrated by the data on ERCC spike-ins, where 80% of read one 5′-ends align to the first base. Strand invasion is inherent in methods using template switching, and we quantify the level of strand invasion to be ~10% in C1 CAGE. For quantifying known TSS, this is manageable as few (~1%) of these strand invasion sites fall within FANTOM5 promoter regions. For identifying de novo CAGE peaks we recommend removing these sites and provide the "findStrandInvaders" function in the CAGEr package. Notably, we could detect the ERCC spike-ins even if they are not capped. Nevertheless, C1 CAGE shows a preference for capped ends, as suggested by the fact that the C1 CAGE library contained only 13% reads from ribosomal RNAs. While this range of ribosomal RNA is acceptable, further reduction might be achieved through the use of pseudo-random primers[51].

The template-switching oligonucleotides (TSOs) included unique molecular identifier (UMIs)[20], however, we have not utilized them for molecular counting, because the TSOs carried over from the reverse-transcription could prime the subsequent

PCR reaction while tolerating mismatches on the UMI sequence, thus causing a high level of mutation rate (as evidenced by the fact that most UMIs are seen only once). Nevertheless, PCR duplicates are partially removed from our data due to the use of paired-end sequencing, as our alignment workflow collapses the pairs that have exactly the same alignment coordinates. Further improvements of the C1 CAGE might address the mutation rate in UMIs. However, in this study, we could identify targets of TGF-β pathway in the absence of UMI and explain the relative and dynamic expression of enhancer RNAs upon stimulation.

Batch effect is a common problem in single-cell RNA-seq, and failing to account for this can lead to cofounding biological interpretations. We introduced, for the first time, an image based approach to decode multiplex samples by using two colors of Calcein AM and their combinations. Moreover, the platform further allows the usage of a larger number of colors or alternatives to Calceins, such as MTT, ATP, or MitoBright, which are generally used for live cell monitoring. For instance, we previously used FUCCI fluorescent reporters to detect cell cycle phases[52]. Other potential applications could include the detection of cytoplasmic or nuclear localizations of fluorescent-labeled transcription factors, or cell division counting with fluorescent probes.

Our cell cycle classification was performed using a model trained on data from H1 hESCs expressing the cell-cycle indicator FUCCI in the C1 system[29]. While training data from phased A549 single-cells would have been preferable, models trained on mouse ESC have also been applied to other cell types with accuracy[30]. However, because the hESC training data was obtained from a 3′-end capture protocol, it may contain different experimental biases that are distinct from our C1 CAGE method. Therefore, these results should be interpreted with caution, and we did not exclude cells based on this classification.

Enhancers have previously been defined by a signature of balanced bidirectional transcription in bulk data[6]. Here, we suggest that this signature arises due to generally mutually exclusive transcription from each strand within single cells. We also suggest for the first time that while eRNAs appear lowly expressed in bulk data, they can be expressed at similar levels to gene promoters within single-cells, although they are expressed in a more restricted subset of cells—i.e., displaying transcriptional bursting.

Notably, C1 CAGE is not restricted to the use in the C1 platform. Indeed, some of the changes introduced in C1 CAGE are also available for bulk nano-CAGE libraries in our latest update[53]. Moreover, the C1 CAGE chemistry might be applicable to profile large numbers of single-cells with droplet-based single-cell capture methods. Droplet technologies are more robust to variations of the cell size, and have higher throughput, although they do not allow for the association of imaging. Five-prime-focused atlases will yield greater insights toward promoter and enhancer activities in various biological systems.

## Methods

**Cell culture of A549 cells and TGF-β stimulation.** A549 cells (ATCC CCL 185) were grown at 37 °C with 5% $CO_2$ in DMEM (Wako, Lot: AWG7009) with 10% fetal bovine serum (Nichirei Bioscience, Lot 1495557) and penicillin/streptomycin (Wako, Lot 168-23191). At 0 h, $10^6$ cells were seeded in three 10 cm dishes (TRP, Cat. num. 93100). At 24 h, the medium was replaced with DMEM without serum after 3 times washing with PBS (Wako, Lot 045-29795). At 48 h, one-third of the dishes were stimulated by treating with 5 ng/ml TGF-β (R&D systems, USA, Accession #P01137). At 66 h, the second third was stimulated with the same treatment. At 72 h, cells for each treatment duration (0, 6, and 24 h) were stained with combinations of Calcein AM and Calcein red-orange (Thermo Fisher Scientific, L3224 and C34851). After 25 min stain at 37 °C, single cell suspensions were prepared by trypsinization and gentle washing for C1 cell loading. Transcriptome alignment of the C1-positive controls against 79 reference genomes of Mycoplasma

or Acholeplasma, including *Mycoplasma hominis*, confirmed the absence of contamination.

**Cell capture.** Calcein stained cells were captured in C1 Single-cell Auto Prep IFC for mRNA Seq, designed for medium-sized (10–17 μm) cells (Cat. Num. 100-5760), following manufacturer's instructions (PN 100-7168). In brief, 60 μl of $2.5 × 10^5$ cell/ml and 40 μl C1 suspension buffer were mixed (all C1 reagents were from Fluidigm), and 20 μl of this mix was loaded into a primed IFC, and processed the script "mRNA Seq: Cell load (1772×/1773×)"

**Imaging.** After loading, IFCs were imaged on INCell Analyzer 6000 (GE Healthcare). Calcein AM was excited at 488 nm and imaged with a FITC fluorescence filter (Semrock). For Calcein red-orange, excitation was at 561 nm (TexasRed; Semrock). Eleven focal planes per chamber and channel were acquired and manually curated to detect empty, dead, singlet, doublet, or multiplet cells in the capture site. In case of single-plane imaging, we used the Cellomics platform like in Böttcher et al.[52] (with a green filter (excitation bandwidth: 480–495 nm, emission bandwidth: 510–545 nm)), and with a red filter (excitation bandwidth: 565–580 nm, emission bandwidth: 610–670 nm (Thermo Scientific)). Processed and raw single-cell images are available for download from http://single-cell.clst.riken.jp/riken_data/A549_TGF___summary_view.php

**Lysis and PCR for C1 CAGE.** Single-cell RNA extraction and cDNA amplification were performed on the C1 IFCs following the C1 CAGE procedure that we deposited in Fluidigm's Script Hub. (https://www.fluidigm.com/c1openapp/scripthub/script/2015-07/c1-cage-1436761405138-3). In brief, cells were loaded in lysis buffer (C1 loading reagent, 0.2% Triton X, 15.2 U Recombinant Ribonuclease Inhibitor, 37.5 pmol reverse-transcription primer, DNA suspension buffer, ERCC RNA Spike-In Mix I or II (Thermo Fisher, 4456653) diluted either 20,000 times (protocol revision B) or 200 times (revision A)), and lysed by heat (72 °C 3 min, 4 °C 10 min, 25 °C 1 min). First-strand cDNAs were reverse transcribed (22 °C 10 min, 42 °C 90 min, 75 °C 15 min) in C1 loading reagent, first-strand buffer, 0.24 pmol dithiothreitol, 15.4 nmol dNTP Mix, betaine, 24.8 U Recombinant Ribonuclease Inhibitor, 175 pmol template-switching oligonucleotide, and 490 U SuperScript III. The cDNAs were amplified by suppressive PCR (which will repress the amplification of the shortest amplicons by the formation of inhibitory "panhandle" structures when the adapters at both ends are complementary) (95 °C 1 min, 30 cycles of 95 °C 15 s, 65 °C 30 s and 68 °C 6 min, 72 °C 10 min) in a mixture containing C1 loading reagent, PCR water, Advantage2 PCR buffer (not SA), dNTP Mix (10 mM each), 24 pmol PCR primer, 50× Advantage2 Polymerase Mix. The PCR products (13 μl) were then harvested in a 96-well plate and quantified with the PicoGreen (Thermo Fisher, P11496) method following the instructions from Fluidigm's C1 mRNA-Seq protocol (PN 100-7168 I1). On-chip cDNA amplification with 30 PCR cycles yielded 1.0 ng/μl in average from single cell. A subset of the samples were further controlled by size profiling on the Agilent Bioanalyzer with High Sensitivity DNA Chip.

**Tagmentation reaction and sequenceing.** Amplified cDNAs were diluted to approximately 0.2 ng/μl following the C1 mRNA-Seq protocol, fragmented, and barcoded by "tagmentation" using the Nextera XT kit (Illumina, cat. num. FC-131-1096-RN) following the instructions from Fluidigm's C1 mRNA-Seq protocol (PN 100-7168 I1), except that we used custom forward PCR primers (dir#501-508/N701-N712, Supplementary Table 3). The final purified library was quality-controlled on a High-Sensitivity DNA Chip and quantified with the KAPA Quantification Kit (Nippon Genetics). Nine pmol were sequenced and demultiplexed on Illumina HiSeq 2500 High output mode (50 nt paired end).

**CAGE processing.** In forward read (Read 1) sequences, linkers were removed and unique molecular identifiers were extracted using TagDust2[54]. Reverse read (Read 2) sequences were then filtered with the program syncpairs (https://github.com/mmendez12/sync_paired_end_reads) to restore the pairing. The pairs were then filtered against the sequences of the human ribosomal RNA locus (GenBank ID U13369.1), and linker oligonucleotides using TagDust2 v2.13 in paired-end mode. They were then aligned to the human genome version hg19 with Burrows Wheeler Aligner (BWA)'s "sampe" method[55] with a maximum insert size of 2,000,000. To map the reads on the ERCC spikes at a single-nucleotide resolution, we prepared reference sequences of the T7 transcription of the ERCC plasmids, which are now available from the NIST's website (https://www-s.nist.gov/srmors/certificates/documents/SRM2374_putative_T7_products_NoPolyA_v1.fasta) (many RNA-seq studies previously published aligned their reads only to the sequence of the plasmid inserts, which lack transcribed linker sequences, which are essential for aligning CAGE reads precisely to the 5′ ends). The properly aligned pairs were then converted to BED12 format with the program pairedBamToBed12 (https://github.com/Population-Transcriptomics/pairedBamToBed12) with the option "-extraG", and assembled in CAGEscan fragments with the program umicountFP (https://github.com/mmendez12/umicount/). This workflow was implemented in the Moirai system (PMID:24884663) and a prototype implemented in a Jupyter notebook is available on GitHub (https://github.com/Population-Transcriptomics/C1-CAGE-preview/blob/master/OP-WORKFLOW-CAGEscan-short-reads-v2.0.

ipynb). The 5′ ends of the CAGEscan fragments represent TSS in the sense of Sequence Ontology's term SO:0000315 ("The first base where RNA polymerase begins to synthesize the RNA transcript").

**Bulk CAGE.** Bulk CAGE data was generated by the nAnT-iCAGE method[27]. Briefly, 5 μg of total RNA prepared from remaining A549 cells after C1 loading. cDNA was reverse transcribed using SuperScript III reverse transcriptase, biotinylated and cap trapped to capture 5' completed cDNAs. Each cDNAs were barcoded and purified. Libraries were sequenced on Illumina HiSeq 2500 high output mode (50 nt single read).

**Image curation and time point demultiplexing.** We used the Bioconductor package CONFESS (v1.6.0)[56] to detect the cells present in the capture chambers, and quantify the fluorescence in the Green and Red channels. In addition, two curators visually screened the images to confirm the presence of cells, and to detect doublets when focal stacks were available. The final annotation reflects the consensus of the three curations. In future studies CONFESS could be used without requiring manual curation. The results were then cross-checked with other quality control parameters, in particular the amount of cDNAs yielded by the C1 runs, and the fraction of spikes and ribosomal RNA in the libraries. In case of conflicting results, chamber images were re-inspected and re-annotated, if necessary.

**ERCC spike-in analysis.** Accuracy and molecular detection limits were calculated as in Svensson et al.[24]: the amount of input spike-in molecules for each spike, for each sample, in each experiment was calculated from the final concentration of ERCC spike-in mix in the sample. The calculation of the accuracy of an individual sample was determined with the Pearson correlation between input concentration of the spike-ins and the measured expression values. Molecular detection limit was calculated with logistic regression using the R function glm from the stats package with family = "binomial".

**Read annotation.** The annotation used combined FANTOM5 robust cage clusters for promoters (http://fantom.gsc.riken.jp/5/datafiles/latest/extra/CAGE_peaks/) and enhancers (http://fantom.gsc.riken.jp/5/datafiles/latest/extra/Enhancers/). Promoter clusters were subtracted from enhancer clusters and annotated to their nearest GENCODEv25 within 500 bp where possible. A mask was added to remove rRNA, tRNA, small RNAs, unannotated promoters. The read 5′ end base only was required to fall into these regions for counting.

**Data processing.** After removing low quality cells and multiple single cells captured sites based on imaging data (SCPortalen[57]), the CAGE reads from the remaining 151 cells that overlapped the annotation CAGE clusters were summed together to create the raw counts matrix. This matrix was processed with the scran package[26] version 1.6.6 in R 3.4.3 for quality control, filtering and normalization. Following the guideline suggested by the authors of scran, we first removed from our analysis 15 cells with (1) library sizes or feature sizes 3 median absolute deviations (MADs) below their median, or (2) mitochondrial proportion or spike proportion 3 MADs above their median, leaving us with 136 cells. All the cells that were dropped due to high-spike proportion also had low-library sizes and feature counts, whereas this was not necessarily true for those that were dropped due to high mitochondrial proportion. Out of 15, 14 removed cells were from the same C1 run (library 2), but there was no noticeable bias towards any particular time point (5, 3, 7 cells from 0, 6, 24 h, respectively). We calculated the cell cycle phase scores using the cyclone method[30] for each cell. We filtered out low-abundance features that were expressed in less than 2 cells or average counts of less than 0.3, leaving us with 18,687 features, of which 826 are FANTOM5 enhancers. These features were normalized with size factors calculated based on clusters of cells with minimum size of 30. We then performed mean-variance trend fitting using the whole endogenous feature set, building the sample replicate and Calcein staining variables into the model. We normalized the expression scores to correct for differences of sequencing depth, using a pooling-deconvolution approach[58]. We then detrended the data for possible C1 run and Calcein color effects. Lastly, we denoised the data by removing low-rank principal components. To produce the final normalized expression levels for downstream analyses, we reduced the technical noise using scran's denoisePCA function based on the fitted data, then performed batch effect removal with the replicate and the Calcein stain as the covariates using limma package's removeBatchEffect function. We selected high variance CAGE clusters (HVCs) as those with biological variation above the 75% quantile and false-discovery rate less than 0.05 after decomposing the total variance for each gene into its biological and technical components using trendVar (scran). We also calculated the pairwise correlations among the HVCs and marked those with FDR greater than 0.05 as significantly correlating HVGs.

To create the pseudotime ordering with TSCAN[28] (version 1.16.0), we selected the input feature set as the union of the significantly correlating HVCs, the top 100 HVCs and SC3[59] defined marker genes, totaling 290 CAGE clusters.

**Comparison to C1 STRT with MESs.** Mouse ESCs (B6G-2) cells purchased from Riken BioResource Center were maintained under feeder-free conditions in

DMEM containing fetal bovine serum (GIBCO), L-glutamine, non-essential amino acids (GIBCO), 2-mercaptoethanol penicillin/streptomycin supplemented with leukemia inhibitory factor. Single-cell suspensions were prepared by accutase for 5 min at 37 °C. After count cells by C-chip (NanoEnTek) and adjustment of cell number, cell capture and live/dead staining were performed by Fluidigm C1 system. Followed by imaging, cDNA synthesis and library prep for C1 CAGE was performed as described above. The final purified library was quality-controlled on a high-sensitivity DNA Chip and quantified with the KAPA Quantification Kit. Totally, 9 pmol were sequenced and demultiplexed on single Illumina HiSeq 2500 Rapid mode (50 nt paired end). C1 STRT data were downloaded (E-MTAB-5482) and split into fastq files for each cell barcode. Read 1 of C1 CAGE data and C1 STRT data were aligned to the mouse genome (mm9) with STAR with parameters (--outFilterMultimapNmax 1 --outFilterScoreMinOverLread 0 --outFilterMatchNminOverLread 0) to select uniquely mapping reads. These reads were then annotated overlapping their 5 prime ends with FANTOM5 CAGE clusters. Strand invasion was calculated for each method using the R package CAGEr function findStrandInvaders with option linker = "GGG".

**Effect of calcein during TGF-β stimulation by qPCR.** Six plates of A549 cells were prepared for three-time points of TGF-β stimulation as described above. In three of six, green and red calcein staining were performed while other three were without calcein staining. Total RNA was extracted from each six plates with RNeasy Mini Kit (QIAGEN). cDNA synthesis was performed by PrimeScript 1st strand cDNA Synthesis Kit (TaKaRa) from 1 μg total RNA. qPCR was performed using SYBR Premix Ex Taq II with ROX Reference Dye (TaKaRa).

**Species-mixing C1 CAGE assay.** Human dermal fibroblast (LONZA, CC-2509) and MEF (MEF_Ng-20D17, Riken BioResource Center) cell lines were used for evaluating the demultiplexing strategy. HDF and MEF cell lines were cultured in DMEM media with 10% fetal bovine serum and penicillin/streptomycin. For validation purpose, two MEF and two HDF plates were prepared for two rounds of C1 CAGE run. First assay was mixed with calcein green stained HDF and both red and green stained MEF. Second assay was mixed with calcein red stained MEF and both green and red stained HDF. After each staining, single-cell suspensions were prepared by trypsinization and washing. After count these cells by C-chip (NanoEnTek), each $1.5 \times 10^5$ cells/ml were mixed in one tube. After cell capture and imaging, two rounds of C1 CAGE reaction were performed. Totally, 192 wells were indexed by index primer, dir #501-516/N701-N712, Supplementary Table 3). The final purified library was quality-controlled on a High-Sensitivity DNA Chip and quantified with the KAPA Quantification Kit. Nine pmol were sequenced and demultiplexed on single Illumina HiSeq 2500 Rapid mode (50 nt paired end). Reads were aligned to a combined mouse (mm10) and human (hg38) genome using the "CAGE processing" pipeline described above, and the fraction of reads uniquely mapping to each genome was calculated.

**Weighted gene co-expression network analysis.** WGCNA[31] version 1.61 was used, with cut height detection threshold of 0.995, minimum module size of 100, signed network type, and merge cut height of 0.25. To reduce noise, we restricted ourselves to those features with mean expression greater than the median of the mean expression across all samples, and biological variation greater than the median. Also, to avoid having the same gene appearing in multiple clusters due to different promoters of the same gene being assigned as such, we only included the major promoter (highest sum of normalized expression across all samples) in the input set, which left us with 6028 CAGE clusters as the input set.

**Motif analysis.** Motif analysis was performed using CAGEd-oPOSSUM[32], which employs two separate scoring systems based on JASPAR 2016[33] transcription factor binding profiles, searching 500 bp either side of CAGE clusters: (1) Z-scores, which counts the total number of a given motif found in the input set, and (2) Fisher score, which counts the number of input regions with the given motif. JASPAR motifs with information content greater than 8 bits were searched.

**Functional analysis.** To see if we could identify any functional characteristics of the genes in each module, we performed a test of gene ontology term over-representation test using the edgeR's goana function, which is an implementation of GOseq[60]. For input, we included those CAGE clusters that showed correlation coefficient of greater than 0.3 with p value less than 0.1 with each module's eigengene. This threshold was chosen by examining the quantile values of the coefficients with p value < 0.1; as the third quantile was 0.29, and 0.3 was chosen for simplicity.

Camera gene set enrichment analysis[45] was performed testing for differential expression between TSCAN states 3 and 4. For the input expression table, we selected the CAGE clusters that were included in the WGCNA analysis and were annotated with Entrezgene IDs. For the test set, we selected those CAGE clusters that showed correlation coefficient of greater than 0.2 with p value less than 0.1 their module's eigengene from the early responders and late responders modules. MSigDB[61] Hallmark gene sets were used.

**Topological association domains**. Out of 826 enhancers, 692 could be assigned to a topological association domain (TAD) identified in A549 cells from ENCODE Dataset GSE105600.

**Fluorescence in situ hybridization**. Enhancer RNA lengths were estimated from the ENCODE A549 RNA-seq signal[62]. We designed oligonucleotide probes consisting of 20 nt targeting sequence using the Stellaris Probe Designer (Biosearch Tech). These sequences were flanked on both ends by 30 nt "readout sequence" serving as annealing sites for secondary probes that are labeled with a fluorescent dye[63]. For each set of probes, all flanking sequences were identical, both on the 5′ and 3′ ends. Positive-strand eRNA, negative-strand eRNA and introns from each locus were assigned different flanking sequences to allow multiplexing. Secondary probes were labeled with either Atto 647 or Cy3 on the 3′ end. All probe sequences are listed in Supplementary Data 2. Briefly, cells were seeded onto coverslips overnight and were fixed in 4% formaldehyde in PBS for 10 min at room temperature. After fixation, the coverslips were treated twice with ice-cold 0.1% sodium borohydride for 5 min at 4 °C. Following three washes in PBS, the coverslips were treated with 0.5% Triton X-100 in PBS for 10 min at room temperature to permeabilize the cells. The coverslips were washed three times in PBS and treated with 70% formamide in 2× SSC for 10 min at room temperature, followed by two washes in ice-cold PBS and another wash in ice-cold 2× SSC. The coverslips were stored at 4 °C for no longer than a few hours prior to hybridization. For hybridization, coverslips were incubated in hybridization buffer containing 252 nM primary probes overnight at 37 °C inside a humid chamber. Hybridization buffer consisted of 10% formamide, 10% dextran sulfate, 2× SSC, 1 μg/μl yeast tRNA, 2 mM vanadyl ribonucleoside complex, and 0.02% BSA. To remove excess probe following hybridization, coverslips were washed twice in wash buffer made of 30% formamide, 2× SSC, 0.1% Triton X-100 for 30 min at room temperature and rinsed once in 2× SSC. For hybridization with secondary probes labeled with fluorescent dyes, coverslips were incubated in minimal hybridization buffer (10% formamide, 10% dextran sulfate, 2× SSC) containing 30 nM secondary probes for 3 h at 37 °C inside a humid chamber. Coverslips were again washed twice in wash buffer for 30 min at room temperature and rinsed once in 2× SSC. Coverslips were mounted on glass slides using ProLong Gold Antifade Mountant with DAPI (Invitrogen). Imaging was done on a DeltaVision Elite microscope (GE) equipped with an Olympus 60× objective (NA 1.42) and a sCMOS camera. Image processing and analysis were done using FIJI.

**Enhancer analysis**. For bidirectionality and epigenetic marks analysis a set of enhancers was selected overlapping ReMap[64] EP300 A549 binding sites. DNase, H3K27ac, H3K4me1, and H3K4me3 bigwig files were downloaded from the NIH roadmap epigenomics project[65] and processed with computeMatrix scale-regions from the deeptools package[66] for enhancer regions. Bidirectional enhancers were selected with at least ten reads in at least five cells and a bidirectionality statistic was calculated as: abs(*plus strand reads—minus strand reads*)/sum(*reads*) ranging from 0 to 1 with 0 being equally bidirectional and 1 being fully unidirectional. Totally, 32 enhancers were selected with absolute score ≤ 0.5. This score was then calculated within each individual cell for these enhancers. The Gini coefficients were calculated using the ineq package in R, using the log expression of the promoters and enhancers calculated by taking the logarithm (base 2) of the raw counts that were incremented by 1. This was done to avoid negative expression values, as Gini coefficient calculation requires positive numbers as input.

**Reporting summary**. Further information on experimental design is available in the Nature Research Reporting Summary linked to this article.

**Code availability**. Code used in this study is available at https://github.com/Population-Transcriptomics/C1-CAGE-manuscript.

## Data availability

C1 CAGE sequence data from this study have been submitted to DDBJ (Project ID: PRJDB5282). Alignments were uploaded to the ZENBU genome browser[67] and a default view is available at http://fantom.gsc.riken.jp/zenbu/gLyphs/#config=NMT9yTLnH59gIVssI9WRfD. In these two submissions the libraries numbered 1, 2, and 3 in this manuscript are numbered 4, 5, and 6, respectively, for historical reasons. The source data underlying all figures are provided as a Source Data file or are present in our code repository.

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

## Acknowledgments

This work was supported by a Research Grant from the Japanese Ministry of Education, Culture, Sports, Science, and Technology (MEXT) to the RIKEN Center for Life Science Technologies. The authors wish to acknowledge RIKEN GeNAS for the sequencing of the libraries, and Fumi Hori for data deposition to DDBJ.

## Author contributions

Conceptualization: T.L., E.A., C.P., and J.W.S. Data curation: T.Ko., A.K., Y.H., M.M., J.Se., I.A., and C.P. Formal analysis: J.M., A.K., Y.H., and E.M. Funding acquisition: P.C. and J.W.S. Investigation: T.Ko., Y.S., S.K., and M.B. Methodology: T.Ko., Y.S., S.K., J.L., C.P., and J.W.S. Project administration: P.C., C.P., and J.W.S. Resources: S.T., T.A., M.F., N.R., J.W., and H.S. Software: J.M., A.K., M.B., M.M., J.Se., I.A., A.H., T.L., and C.P. Supervision: H.S., T.Ka., T.L., C.C.H., E.A., C.P., and J.W.S. Validation: T.Ko. and Y.S. Visualization: J.M., A.K., I.A., and C.P. Writing—original draft: T.Ko., J.M., A.K., Y.S., E.A., C.P., and J.W.S. Writing: review and editing: J.M., C.P., and J.W.S.

## Additional information

**Competing interests:** N.R. is an employee and stockholder of Fluidigm Corporation. The remaining authors declare no competing interests.

