## [Peer Review File · Nature Communications]

Reviewers' comments:

Reviewer #1 (Remarks to the Author):

Kouno et al. describe a new method for 5'-specific sequencing of transcripts at single-cell resolution. They validate their method on ERCC data, and show that it can be used to resolve biological differences along a EMT trajectory. They describe results relating to the unidirectionality of eRNA transcription in single cells, as well as the specificity of eRNA expression compared to expression from promoters. They also present a multiplexing strategy based on colored stains. The manuscript is well-written and clear, and the methodology and the results are interesting. However, I have some concerns about both the experimental validation and the computational analysis, which are described in more detail below.

MAJOR:

1. The presence of strand invasion events is concerning for a method that claims to achieve 5'-specific coverage. This will reduce the usefulness of the method for detecting de-novo TSSs, as one cannot be sure whether a peak in coverage arises from genuine transcription or due to these artefacts. For quantifying transcription around known TSSs, I imagine that this is less of a problem as artefacts distant from an annotated TSS can be simply ignored. I would like to see some more discussion of this; in particular, on the "tolerance" with which reads are assigned to annotated promoter/enhancers, as a low tolerance may be able to avoid many problems with the non-specific artefacts.
2. The failure of the UMIs is unfortunate, as I am not convinced by the current strategy for removing duplicates. While it is true that the data are paired-end, only one end is actually free to vary as the other is anchored to the 5' end of the transcript. This means that duplicates will be incorrectly detected at high-coverage sites - a problem endemic to single-end CHIP-seq data, for example. It would be a great improvement to the protocol if UMI information could be successfully incorporated, e.g., by adding UMIs to the random primer instead.
3. The color-based multiplexing strategy is interesting and likely to be generally useful. However, I would like more evidence that the procedure is working correctly.
 - Specifically, I would like to see a control experiment where different known cell types (ideally from different species) are labelled with different colors. The aim would be to demonstrate that the de-multiplexing correctly separates the known groups. This would show that there are no issues from, e.g., loss of label from a cell (potentially causing a yellow cell to be called as a red or green cell) or uptake of a different label during processing.
 - I would also like to see a demonstration that the introduction of the label has no effects on transcription. This would be fairly easy to achieve by showing that there are no DE genes after treatment of cells with the label.
4. The text at the end of page 6 describes a bulk experiment, but does not show the results of the comparison. I can only find a short comment on page 10 regarding the use of the bulk data. I would like to see much more extensive comparisons between the single-cell and bulk data, e.g., by comparing the number and coverage of CAGE clusters in the pooled single-cell data with that in the bulk.
5. The eRNA directionality analysis does not account for the fact that the coverage in single cells is lower than that of the pool. With lower coverage, the directionality statistic would naturally be larger in magnitude. For example, if an enhancer locus only had 1 read in a single cell, the

directionality must always be 1. Thus, the differences between the pooled and single-cell mean in Figure 4C are likely to be misleading. To fix this, I suggest recomputing the statistics after downsampling the pooled counts to the same coverage as observed in the single cells. This will account for the differences in coverage and improve the validity of the comparison of directionality between pooled and single cells.

6. Assuming that the enhancers are indeed unidirectionally transcribed in individual cells, I have some further questions:

- Only 32 enhancers (with balanced transcription in the pool) were examined. There must be thousands of enhancers in the mammalian genome - what is the biological significance of unidirectional transcription for a tiny subset of enhancers?
- Is the unidirectionality related to the timepoint or TSCAN state?
- Through what putative mechanism does this unidirectionality occur?

7. The FISH validation in Figure 5 is not convincing. The association between the FISH and CAGE data in Figures 5A and 5B is not strong. Is this due to an inherent discrepancy between protocols, or is there something wrong with the single-cell CAGE? This would be detectable by examining the coverage in bulk data and comparing that to the coverage in C1 CAGE and the total number of detected transcripts from smFISH.

8. From colocalization in the FISH data, the authors suggest that the enhancer is involved in regulating nearby promoters (page 15). Does smFISH have sufficient resolution to make that claim? It seems inevitable that fluorescent spots at two adjacent loci would overlap, especially given that both spots seem to occupy a fairly large area in Figure 5D. However, this may not correspond to a physical interaction - that would require chromatin conformation capture experiments to prove.

MINOR:

- I would like to see plots of the observed and expected ERCC concentrations to supplement Figure 1C.

- I assume that the yellow label in Figure 2A was a mix of red and green from cells that were co-stained with both - is this the case?

- I feel that the WGCNA results are unnecessarily complicated for the question being asked, which is "how do the cells change along the TSCAN trajectory?" A more direct approach would be to detect DE genes at each TSCAN stage against the previous stage, which would identify the transcriptional changes at each segment of the trajectory. The authors should consider using this approach instead, which would make it easier to interpret the results; avoid problems with "weak responders", which are clearly not responding at all; and provide more statistical rigour, in contrast to arbitrary thresholds of 0.3 for the correlation and 0.1 for the p-value (without any apparent correction for multiple testing).

- I am not sure what "bi-phasic" means on page 11, this could be clarified.

- There is no need to devise a new specificity score on page 13; the Gini index is a well-established measure for this purpose.

- The justification for the use of gene-based size factors is not quite correct. The real choice is whether or not total RNA content is of interest, as discussed by Lun et al. (2017) (10.1101/gr.222877.117).

- Can C1 CAGE be used with microwell plates?

- In the Methods section, it is mentioned that three curations were performed during demultiplexing. Is this really necessary? Requiring three separate curations (or any manual curation at all) is unlikely to be scalable for thousands of cells.

- What was the exact procedure used to assign reads to promoters or enhancers? For example, were reads only assigned if the 5' end of the read was on the same base (or plus/minus some tolerance) as the promoter/enhancer region?

- The description of the molecular detection limit calculation on page 21 is insufficient. There are many ways to use glm(); I presume that the authors performed some kind of logistic regression on the presence/absence of the ERCC in each cell with respect to abundance? This should be clarified.

- In the single-cell data analysis on page 22, the batch correction seems to be performed after PCA. This should be the other way around, otherwise the first few PCs will be dominated by the batch effect.

- There is no description of the expression values used as input to TSCAN. Is it the log-expression values? Denoised log-expression values? The PCs from denoisePCA?

Reviewer #2 (Remarks to the Author):

In this paper, Tsukasa Kouno et al have developed a new single-cell RNA-seq method called C1 CAGE, based on the nanoCAGE method. In brief, the method used random hexamers for reverse transcription, followed by template switching and pre-amplification. Then cDNA is tagged by Tn5 and the 5'-end of cDNA is specifically amplified by index PCR. The method is adapted to the C1TM Single-Cell Auto Prep System (Fluidigm). Using this method, the authors profile promoter and enhancer activities in the cellular response to TGF- β of lung cancer cells and showed transcriptional bursts in subsets of cells with transcripts arising from either strand within a single-cell in a mutually exclusive manner.

In my opinion, the principle of the method complements the present single cell RNA-seq methods and thus this study should be a useful contribution to the single-cell transcriptomics field. However, some major and minor concerns should be addressed.

Major:

1) One of my major concerns is that the detected transcripts per cell of C1 CAGE seems to be low (2,788 CAGE clusters detected per cell). Regarding that the total 18,687 CAGE clusters covered 9,809 genes, it seems that the method only detected about 1500 genes per cell. Low detection sensitivity will greatly limit the application of the method. The authors have compared with the C1 STRT Seq for ERCC detection. Since the ERCC spike-in RNAs have short polyA tail about 20nt, this comparison indeed show that C1 CAGE works better than STRT Seq for non-polyA RNA detection, but is not enough to fully assess the detection sensitivity of the method. The authors should directly compare the detection sensitivity including the detected transcripts per cell between this and existing single cell RNA-seq methods using the same cell type or purified RNA. Calculating the detection efficiency by comparing with smFISH is optional.

2) More experiments are needed to support the conclusion "transcripts arising from either strand within a single-cell in a mutually exclusive manner". The authors calculated the bidirectionality score between the single cell mean results and the pooled results. The authors also selected enhancers

with relatively abundant reads trying to avoid dropout bias. However, the filter condition of "at least 10 reads in at least 5 cells" may not be enough to avoid dropout bias. To prove this, the authors should perform the pool-and split experiment. For example, 100 cells or equivalent RNA are pooled together and split into 100 unit for C1 CAGE, to test if the bidirectionality score is significantly lower in the pool-and-split experiment than in the real single cell experiment.

3) The smFISH experiment targeting the + and – strands of the eRNA is important to validate the conclusion "transcripts arising from either strand within a single-cell in a mutually exclusive manner". However, the results seems to display that both strands of KLF6-eRNA1 are expressed in an individual cell (the middle panel of Figure S8b), while the percentage of the cells expressing both eRNA strands is not shown (the left panel of Figure S8b). These data should be presented more clearly. Also, I think these data should be put into the formal instead of supplementary figure due to their importance to one main conclusion.

Minor:

- 1) The rRNA transcripts only present at a percentage of 13% in C1 CAGE data. Can the authors explain how C1 CAGE avoids amplifying rRNA, regarding that the used random hexamers will also transcript rRNA?
- 2) Will C1 CAGE also amplify genomic DNA? Can the authors assess the percentage of genomic DNA amplified reads in the C1 CAGE data?
- 3) The authors showed that most ERCC RNA detections are located at 5' end. Can the authors also show the situation of real cellular mRNAs?
- 4) Can the author explain the "suppressive PCR" in Figure 1A?

Reviewers' comments:

Reviewer #1 (Remarks to the Author):

Kouno et al. describe a new method for 5'-specific sequencing of transcripts at single-cell resolution. They validate their method on ERCC data, and show that it can be used to resolve biological differences along a EMT trajectory. They describe results relating to the unidirectionality of eRNA transcription in single cells, as well as the specificity of eRNA expression compared to expression from promoters. They also present a multiplexing strategy based on colored stains.

The manuscript is well-written and clear, and the methodology and the results are interesting. However, I have some concerns about both the experimental validation and the computational analysis, which are described in more detail below.

MAJOR:

1. The presence of strand invasion events is concerning for a method that claims to achieve 5'-specific coverage. This will reduce the usefulness of the method for detecting de-novo TSSs, as one cannot be sure whether a peak in coverage arises from genuine transcription or due to these artefacts. For quantifying transcription around known TSSs, I imagine that this is less of a problem as artefacts distant from an annotated TSS can be simply ignored. I would like to see some more discussion of this; in particular, on the "tolerance" with which reads are assigned to annotated promoter/enhancers, as a low tolerance may be able to avoid many problems with the non-specific artefacts.

We thank the reviewer for raising this important issue. We agree that strand invasion is concerning. While these artefacts are found in all methods based on the template-switching reaction (e.g. STRT, Smart-Seq), we chose to disclose C1 CAGE's biases in the manuscript which should increase confidence and not deter users. When compared to C1 STRT, we observed a higher rate of strand invasion than C1 CAGE (Figure S1a). We also note that these artefacts are created during reverse-transcription (Tang et al., 2013), which means that they only arise from expressed genes, that is, "genuine transcription".

As regards to the reviewer's comment on "tolerance", we used the reference FANTOM5 CAGE peaks as a source of known TSS in this manuscript and did not filter out the artefacts since this is a very stringent approach (<1% of strand invaders overlap TSS). In fact, we recommend to keep the artefacts in the data when intersecting to broader promoter regions (such as TSS +/- 500 nt), or when quantifying the activity of whole genes (such as assigning expression counts to gene symbols), because the main effect of artefact formation is to shift some CAGE signal downstream the real TSS. Conversely, we advise to remove the artefacts when performing *de novo* TSS peak calling. We provide the findStrandInvaders function from the CAGER package (version > 1.22) which can remove strand invaders, where the tolerance threshold can be freely adjusted to the needs of the analysis. We updated this point in the revised version of the manuscript under discussion (lines 390-394).

2. The failure of the UMIs is unfortunate, as I am not convinced by the current strategy for removing duplicates. While it is true that the data are paired-end, only one end is actually free to vary as the other is anchored to the 5' end of the transcript. This means that duplicates will be incorrectly detected at high-coverage sites - a problem endemic to single-end ChIP-seq data, for example. It would be a great improvement to the protocol if UMI information could be successfully incorporated, e.g., by adding UMIs to the random primer instead.

We agree with the reviewer that molecular counting with UMIs would be highly desirable. However, reviewer's suggestion of adding UMIs to the random primer is not compatible with the current design of the C1 CAGE protocol, because the custom PCR that follows the tagmentation reaction discards the 3' ends of the molecules. Our standpoint for this manuscript is that the core validation experiment (the TGF- β timecourse), which has been done with the UMI-absent C1 CAGE protocol, has reached its goal of identifying targets of the TGF- β pathway and cannot be redone at a reasonable time and cost. However, we agree with the reviewer that PCR duplicate removal can lead to underestimation of expression scores. On the other hand, lack of removal can also lead to distortion of expression scores, and it is difficult to decide which option is the best. We chose to remove duplicates, also

because it makes the subsequent steps of the computation lighter, but other researchers can easily choose to skip that step. In addition, during tagmentation, a single molecule can generate an arbitrary number of fragments with different 3' ends, and we assume that this will milder the possibility of under-estimating expression counts because of cleaving two molecules at the same coordinates is rare. Lastly, while the 5' end is anchored, it can still vary on a scale of a few bases to hundred bases, in case of broad promoters, which again reduces the possibility of producing the same coordinate pairs from different transcript molecules.

3. The color-based multiplexing strategy is interesting and likely to be generally useful. However, I would like more evidence that the procedure is working correctly.

- Specifically, I would like to see a control experiment where different known cell types (ideally from different species) are labelled with different colors. The aim would be to demonstrate that the de-multiplexing correctly separates the known groups. This would show that there are no issues from, e.g., loss of label from a cell (potentially causing a yellow cell to be called as a red or green cell) or uptake of a different label during processing.

We thank the reviewer for raising this concern. To directly address this issue, we performed a new species mixing experiment with two C1 CAGE runs mixing human dermal fibroblasts and mouse embryonic fibroblasts labeled with different combinations of colors. Aligning reads from these experiments to the combined human and mouse genomes, we observe complete agreement between the cell assignment from fluorescent label and the genome to which the majority of uniquely mapping reads are aligned (Figure S4), suggesting that the cell labeling by Calcein AM does not affect sample de-multiplexing. We have updated Figure S3 and the text in lines 151-155.

- I would also like to see a demonstration that the introduction of the label has no effects on transcription. This would be fairly easy to achieve by showing that there are no DE genes after treatment of cells with the label.

We thank the reviewer for this point. Calcein AM is a widely used dye for determining cell viability currently recommended by Fluidigm C1 in their LIVE/Dead staining kit (2 μ M Calcein AM at final concentration) in all C1 runs to distinguish live and dead cells

To minimize the unwanted effect of cell staining in our experiment, we performed 25 min staining at 37°C under native medium, and decreased Calcein AM to 1 μ M at final concentration (half the suggested concentration). To further check the effect on gene expression, we compared six TGF- β response genes after Calcein AM staining (Green and Red) in TGF- β stimulated A549 cells by qPCR. The expression levels had minor or non-significant changes and exhibited the same expression patterns during TGF- β stimulation as observed in the C1 CAGE result (Figure S3).
Updated: Figure S2 and text at lines 151-152

4. The text at the end of page 6 describes a bulk experiment, but does not show the results of the comparison. I can only find is a short comment on page 10 regarding the use of the bulk data. I would like to see much more extensive comparisons between the single-cell and bulk data, e.g., by comparing the number and coverage of CAGE clusters in the pooled single-cell data with that in the bulk.

We thank the reviewer for the suggestion. Figure 4e shows the relationship between bulk and single cell coverage in the original manuscript however we agree with the reviewer that it was insufficient to address the sensitivity of the single cell method. To address this, we additionally analyzed number and coverage of CAGE clusters in the pooled single cell data with that in the bulk. Bulk CAGE experiments detect ~13,000 protein coding genes whereas each single cell detects on average 2,668. However pooling single cells, we could observe 10,822 protein coding genes. We included more discussion at lines 177-179 and introduce Figure S6 to compare bulk and single cell methods in the revised version of this manuscript.

5. The eRNA directionality analysis does not account for the fact that the coverage in single cells is lower than that of the pool. With lower coverage, the directionality statistic would naturally be larger in magnitude. For example, if an enhancer locus only had 1 read in a single cell, the directionality must always be 1. Thus, the differences between the pooled and single-cell mean in Figure 4C are likely to be misleading. To fix this, I suggest recomputing the statistics after downsampling the pooled counts to the same coverage as observed in the single cells. This will account for the differences in coverage and improve the validity of the comparison of directionality between pooled and single cells.

We thank the reviewer for raising this important point, as this question was also raised by the other reviewer. In this study, we defined 'robust' enhancers to have minimum reads (10) and cells (5) criteria to increase the chance of detecting real eRNAs, however the pooled counts were not corrected for sequencing depth. To address this point, we have downsampled 100 times for each enhancer reads from the pooled cells to the same depth as their corresponding single cells and show the resulting mean and standard deviations. Notably, these reads also show a bidirectional signature, in contrast to the single cells, suggesting that eRNAs are largely unidirectional at the single cell resolution. We have revised Figure 4c to show this.

6. Assuming that the enhancers are indeed unidirectionally transcribed in individual cells, I have some further questions:

- Only 32 enhancers (with balanced transcription in the pool) were examined. There must be thousands of enhancers in the mammalian genome - what is the biological significance of unidirectional transcription for a tiny subset of enhancers? Is the unidirectionality related to the timepoint or TSCAN state?

Thank you for this question. For clarity, only 32 robust enhancers with at least 10 reads in 5 cells and balanced bidirectional transcription in pooled cells were selected for the analysis. To show the distribution of eRNAs in our dataset, we added supplementary figure 11 with the number of enhancers found at various cell and read count thresholds, with and without requiring bidirectional transcription. We have updated lines 279-284.

In general, we expect more enhancers with unidirectionality to exist in the genome; not only 32 out of thousands of enhancers, as mentioned by the reviewer. Possible biological explanation for unidirectionality may be due to steric occupancy of RNA PolII on each strand at any given time, or regions proximal to chromatin loop structures such as CTCF. However, given relatively low/rare expression in our data, it is hard to draw conclusions about the biological meaning of directionality. As this relates to timepoint or TSCAN state; we could not observe time dependent effect when cells were ordered according to pseudotime or TSCAN states. Replotted data for example eRNA in Figure 4d can be found in supplementary figure 12.

7. The FISH validation in Figure 5 is not convincing. The association between the FISH and CAGE data in Figures 5A and 5B is not strong. Is this due to an inherent discrepancy between protocols, or is there something wrong with the single-cell CAGE? This would be detectable by examining the coverage in bulk data and comparing that to the coverage in C1 CAGE and the total number of detected transcripts from smFISH.

We appreciate the comments from the reviewer. We note the discrepancy observed between FISH and C1 CAGE. In Figure 5a and b, smFISH and C1 CAGE show similar trends of expression of KLF6-eRNA1. However, the percentage of cells in which signal is detected is different between the two methods. This is expected due to discrepancy between the methods, as the reviewer pointed out, leading to sensitivity differences. Single cell RNA sequencing technologies generally suffer from dropout rates as we observe in our comparison to bulk CAGE (Fig 4e). When compared to bulk CAGE expression (Figure S14), we observed similar patterns as C1 CAGE at these loci, indicating cumulative expression of eRNAs.

Visualization by tiling of fluorescently-labeled probes along the target transcript (having good signal-to-noise ratio), as in the case of smFISH, would be more sensitive and detect a higher number of cells possessing the eRNA of interest. Similar trend between the two methods, however, serves as validation for the ability of C1 CAGE to detect dynamic expression of eRNAs.

8. From colocalization in the FISH data, the authors suggest that the enhancer is involved in regulating nearby promoters (page 15). Does smFISH have sufficient resolution to make that claim? It seems inevitable that fluorescent spots at two adjacent loci would overlap, especially given that both spots seem to occupy a fairly large area in Figure 5D. However, this may not correspond to a physical interaction - that would require chromatin conformation capture experiments to prove.

The reviewer is justified to raise the resolution issue. Although the KLF6 gene and its supposed enhancer are approximately 100 kb apart, which linearly (in 1D) may occupy a distance of more than 30 μm (assuming 0.33 μm per kb of DNA, as shown in Dickerson et al (1982) Science, 216: 475-485), their three-dimensional distance in the nucleus would be much less, especially as the two are in the same TAD which would be characterized by multiple long-range contacts. On the other hand, linear distance between the PMEPA1 gene and its nearby enhancer is roughly 7 kb; the three-dimensional distance will be even shorter. The resolution of FISH in the 3D context of the nucleus would be locus-dependent, as it would depend on the local chromatin conformation. As a rough guideline, a single pixel in these images is 108 nm (in both x and y dimensions). Distance between the centers of two co-localizing FISH spots (occurring on the same z-plane) in these images occupy 5 pixels (540nm) or more, which may be much greater than the effective sizes of the molecules concerned (eRNA and TSS on target DNA). It is therefore not our intention to over-claim the physical association between the eRNA and its supposed target gene based on FISH spot localization alone. Rather, the FISH spot co-localization suggests that the eRNA and KLF6 are co-expressed on the same allele. We take advantage of the fact that co-expression from the same allele inevitably leads to co-localized spots. Since both the eRNA and KLF6 intron are in the same TAD and show similar trends in expression from the same allele following stimulation, we are raising the possibility that this eRNA may be associated with KLF6. To clarify this issue, we have modified the manuscript at lines 359-361.

MINOR:

- I would like to see plots of the observed and expected ERCC concentrations to supplement Figure 1C.

Figure S1b,c are added to show the expected and observed ERCC molecules across all samples in C1 CAGE and C1 STRT.

- I assume that the yellow label in Figure 2A was a mix of red and green from cells that were co-stained with both - is this the case?

Yes, this is correct. The figure legend has been updated to clarify this point

- I feel that the WGCNA results are unnecessarily complicated for the question being asked, which is "how do the cells change along the TSCAN trajectory?" A more direct approach would be to detect DE genes at each TSCAN stage against the previous stage, which would identify the transcriptional changes at each segment of the trajectory. The authors should consider using this approach instead, which would make it easier to interpret the results; avoid problems with "weak responders", which are clearly not responding at all; and provide more statistical rigor, in contrast to arbitrary thresholds of 0.3 for the correlation and 0.1 for the p-value (without any apparent correction for multiple testing).

We thank the reviewer for the comment. With respect to weak responders we agree that they are mostly non-responding, as we previously stated in the text: "whereas Weak Responding I and II represent those with little or no changes in their transcription", and we focus our analysis on the other modules.

As for the thresholds, we chose a lenient cutoff for defining module membership to reduce the noise in the dataset for more functional downstream analyses.

Performing differential expression analysis between subsequent time points would be limited, as it necessarily identifies all the genes that undergo expression changes at the time point being compared, irrespective of their behaviours at other time points. As such, one cannot assign module memberships to individual genes based on their overall behaviour over the entire expression profile (e.g. over the course of the TGFbeta activation process). The advantage of WGCNA is that it can cluster the genes based on their overall expression profile across all time points or pseudotime. This is important for our functional analysis into the Early and Late Responders, as we needed to restrict ourselves to only those genes that show much stronger differential expression in the final stage of TGFbeta response compared to other stages.

- I am not sure what "bi-phasic" means on page 11, this could be clarified.

We used 'Bi-phasic' to mean two cellular states at a given time point. To clarify this, we modified the manuscript to remove the word 'bi-phasic' and explicitly state this at line 266.

- There is no need to devise a new specificity score on page 13; the Gini index is a well-established measure for this purpose.

We would like to thank the reviewer for suggesting the Gini index, which is widely used across different fields. Instead of devising our own custom methodology, we have re-evaluated the difference between promoters and enhancers using this index observing similar results, and revised the manuscript accordingly (Figure 4g, lines 309-314).

- The justification for the use of gene-based size factors is not quite correct. The real choice is whether or not total RNA content is of interest, as discussed by Lun et al. (2017) (10.1101/gr.222877.117).

We thank the reviewer for pointing out our incorrect justification. We have simplified this section of the discussion and details of the normalization are found in the methods section lines 580-590.

- Can C1 CAGE be used with microwell plates?

C1 CAGE is based on nanoCAGE, and we have been using nanoCAGE on single-cells in microplates in a work that is still unpublished. It would be probably possible to run C1 CAGE in microplates, but we are not mentioning that fact in the manuscript because we have not tried and will not be able to support third parties trying this.

- In the Methods section, it is mentioned that three curations were performed during demultiplexing. Is this really necessary? Requiring three separate curations (or any manual curation at all) is unlikely to be scalable for thousands of cells.

Manual curation was performed as proof-of-principle to demonstrate robustness of the method. Indeed, the number of curations reflects that the data presented was produced during the development of the method. For new projects we would recommend to automate color demultiplexing using the CONFESS software package available on Bioconductor: <https://www.bioconductor.org/packages/release/bioc/html/CONFESS.html>. The text has been updated at line 546 to state this.

- What was the exact procedure used to assign reads to promoters or enhancers? For example, were reads only assigned if the 5' end of the read was on the same base (or plus/minus some tolerance) as the promoter/enhancer region?

We required a strict match between the first base of the extracted read and the FANTOM5 TSS region. We curated the FANTOM5 regions so that no enhancers overlap promoter regions. This is described in the methods subsection 'read annotation', which we have updated for clarity (line 567).

- The description of the molecular detection limit calculation on page 21 is insufficient. There

are many ways to use glm(); I presume that the authors performed some kind of logistic regression on the presence/absence of the ERCC in each cell with respect to abundance? This should be clarified.

We implemented the method used by (Svensson et al., 2017) using logistic regression and our source code is available at in our github repository <https://github.com/Population-Transcriptomics/C1-CAGE-manuscript> with the code: `glm(detected ~ mol, data = df, family = binomial)`, where `detected = {0,1}`, `mol = input molecules`. The manuscript at line 558 has been updated to add this.

- In the single-cell data analysis on page 22, the batch correction seems to be performed after PCA. This should be the other way around, otherwise the first few PCs will be dominated by the batch effect.

- There is no description of the expression values used as input to TSCAN. Is it the log-expression values? Denoised log-expression values? The PCs from denoisePCA?

Thank you for raising these comments and we now understand the ambiguity in the order of batch correction. To clarify, there are two sets of PCA plots generated in the manuscript:

- 1) Figure S1 is used to illustrate the effect of batch correction: (a) uncorrected, and (b) batch-corrected. These plots are constructed using the uncorrected and batch-corrected expression tables, respectively, from the full set of 18,687 CAGE clusters.
- 2) Figure 2 (d,e) are based on the normalized, batch-corrected log₂ expression values for the variable subset of CAGE clusters used as input for TSCAN.

To make things clearer, we added a sentence on page 6, line 174: "...All subsequent analyses are based on this normalized and batch-corrected expression table in log₂ scale..."

Reviewer #2 (Remarks to the Author):

In this paper, Tsukasa Kouno et al have developed a new single-cell RNA-seq method called C1 CAGE, based on the nanoCAGE method. In brief, the method used random hexamers for reverse transcription, followed by template switching and pre-amplification. Then cDNA is tagged by Tn5 and the 5'-end of cDNA is specifically amplified by index PCR. The method is adapted to the C1TM Single-Cell Auto Prep System (Fluidigm). Using this method, the authors profile promoter and enhancer activities in the cellular response to TGF- β of lung cancer cells and showed transcriptional bursts in subsets of cells with transcripts arising from either strand within a single-cell in a mutually exclusive manner.

In my opinion, the principle of the method complements the present single cell RNA-seq methods and thus this study should be a useful contribution to the single-cell transcriptomics field. However, some major and minor concerns should be addressed.

Major:

1) One of my major concerns is that the detected transcripts per cell of C1 CAGE seems to be low (2,788 CAGE clusters detected per cell). Regarding that the total 18,687 CAGE clusters covered 9,809 genes, it seems that the method only detected about 1500 genes per cell. Low detection sensitivity will greatly limit the application of the method. The authors have compared with the C1 STRT Seq for ERCC detection. Since the ERCC spike-in RNAs have short polyA tail about 20nt, this comparison indeed show that C1 CAGE works better than STRT Seq for non-polyA RNA detection, but is not enough to fully assess the detection sensitivity of the method. The authors should directly compare the detection sensitivity including the detected transcripts per cell between this and existing single cell RNA-seq methods using the same cell type or purified RNA. Calculating the detection efficiency by comparing with smFISH is optional.

We thank the reviewer for raising this important issue. In the revised manuscript, we made additional comparisons with C1 STRT. It is notable to state that C1 STRT requires custom transposase mixes that are not commercially available, therefore running a side-to-side comparison in our laboratory would require an open-ended amount of work that would considerably delay the resubmission of this manuscript.

However, to address reviewer's comment, we performed a C1 CAGE experiment on mouse ES cells to allow comparison with a previously generated C1 STRT dataset. We down-sample both datasets to 500,000 reads per cell and observe that C1 CAGE identifies a median 2,991 protein coding genes per cell, whereas C1 STRT identifies median 2,335. Distributions are shown in Figure 1f of the revised version of the manuscript. This indicates that C1 CAGE is as sensitive or possibly more in detecting genes using the same cell type.

2) More experiments are needed to support the conclusion "transcripts arising from either strand within a single-cell in a mutually exclusive manner". The authors calculated the bidirectionality score between the single cell mean results and the pooled results. The authors also selected enhancers with relatively abundant reads trying to avoid dropout bias. However, the filter condition of "at least 10 reads in at least 5 cells" may not be enough to avoid dropout bias. To prove this, the authors should perform the pool-and split experiment. For example, 100 cells or equivalent RNA are pooled together and split into 100 unit for C1 CAGE, to test if the bidirectionality score is significantly lower in the pool-and-split experiment than in the real single cell experiment.

We thank the reviewer for this comment. Indeed, this is an important issue which was also raised by reviewer one. We believe that the pool-and-split experiment can be recapitulated by down-sampling pooled single cell data. In practice, we have down-sampled 100 times for each enhancer reads from the pooled single cells to the same depth as their corresponding single cells and show the resulting mean and standard deviations. Notably, these reads also show a bidirectional signature, in contrast to the single cells, suggesting that eRNAs are largely unidirectional at the single cell resolution. To clarify this point, we have revised Figure 4c to show this.

3) The smFISH experiment targeting the + and – strands of the eRNA is important to validate the conclusion "transcripts arising from either strand within a single-cell in a mutually exclusive manner". However, the results seems to display that both strands of KLF6-eRNA1 are expressed in an individual cell (the middle panel of Figure S8b), while the percentage of the cells expressing both eRNA strands is not shown (the left panel of Figure S8b). These data should be presented more clearly. Also, I think these data should be put into the formal instead of supplementary figure due to their importance to one main conclusion.

Thank you for the comment. We regret misleading the reviewer with a FISH image that was not representative. The image indeed showed eRNA from both strands on the same allele, which is actually a rare event. We placed this image only to show that we are capable of detecting both strands from the same allele, should they occur. We have replaced this image with a more representative one, showing expression from a single strand only. The previous image showing both strands still remains in the figure, as it is important to show that bidirectional transcription from the same allele, although rare, does occur and visualizing this in a FISH image is very convincing. The reviewer is also right to mention that the frequency of cells expressing both strands is not shown in the figure. We have now added a new panel in Figure S15 to show this.

Minor:

1) The rRNA transcripts only present at a percentage of 13% in C1 CAGE data. Can the authors explain how C1 CAGE avoids amplifying rRNA, regarding that the used random hexamers will also transcript rRNA?

We were also surprised by this result. One possible interpretation could be that some of the rRNA stays bound to the ribosomes after cell lysis, and therefore does not contribute to the libraries. Another possibility could be that the nuclei, which also contain large amounts of rRNA, are only partially lysed. However, this is contradicted by our observation of nuclear transcripts such as MALAT 1.

2) Will C1 CAGE also amplify genomic DNA? Can the authors assess the percentage of genomic DNA amplified reads in the C1 CAGE data?

From a mechanistic point of view, the barriers to the incorporation of genomic DNA in the libraries are that 1) the DNA should be single-stranded in order to be primed and extended by the reverse-transcriptase and 2) the genomic DNA molecules do not have a cap and will be disfavored by the template switching reaction. Thus, we do not expect to see much contamination from the genomic DNA. Furthermore, given the pervasive nature of transcription, it is hard to rule out that whether a CAGE pair aligned in a genomic region did not originate from an RNA molecule.

3) The authors showed that most ERCC RNA detections are located at 5' end. Can the authors also show the situation of real cellular mRNAs?

This has been added to supplementary figure 6 to show the distribution of 5' ends in C1 CAGE and bulk CAGE.

4) Can the author explain the "suppressive PCR" in Figure 1A?

Suppressive PCR will repress the amplification of the shortest amplicons by the formation of inhibitory "panhandle" structures when the adapters at both ends are complementary. See for instance Siebert et al., 1995 (<https://www.ncbi.nlm.nih.gov/pubmed/7731798>). This has been added to the methods section line 488.

REVIEWERS' COMMENTS:

Reviewer #1 (Remarks to the Author):

The revisions performed by the authors have addressed my concerns.

Reviewer #2 (Remarks to the Author):

My questions have been answered.